# The Impact of the Great Recession on Well-Being across Europe Ten Years On: A Cluster Analysis

**Elisabetta Croci Angelini** [1,*] **, Francesco Farina** [2] **and Silvia Sorana** [3]

1 Center for Interdisciplinary Studies in Economics, Psychology and Social Sciences (CISEPS), University of Milano-Bicocca, 20126 Milan, Italy
2 Centre for Investigation and Modelling of Experimental Observations (CIMEO), Sapienza University of Rome, 00161 Rome, Italy
3 COOSS Marche Onlus, 60121 Ancona, Italy
* Correspondence: elisabettacroci@gmail.com

**Abstract:** To evaluate variations in the well-being dimensions of European citizens, we rely upon Principal Component Analysis methodology, whereby a large set of interrelated indicators are reduced to a small number of aggregate synthetic variables. We find that the 2008 crisis impinged differently on the various dimensions of well-being. The evolution of the indicators has affected different clusters of countries in various ways. Most importantly, we observe that there has been a shift of the principal component from the poor in terms of material deprivation to the risk of poverty for the worsening conditions in the labor market.

**Keywords:** European Union; Great Recession; well-being; Principal Component Analysis

## 1. Introduction

The Great Recession had widespread consequences. During the 2008 financial crisis the interplay between the mutual exposure of banks and governments to the other party's insolvency risk greatly distressed the balance sheets of the banks, finally leading to a credit crunch. A severe recession in advanced countries, with rising unemployment and negative growth rates, caused a lower demand level. In Europe, diminishing earnings for households and declining profits for firms coupled with the slackened functioning of automatic stabilizers posited the Eurozone GDP dynamics on a lower path (De Grauwe and Ji 2013). To counter rising public deficit and public debts with respect to GDP, austerity policies were implemented through restrictive impulses of fiscal policy.

The less efficient countries participating in the European Monetary Union have been exposed to the divergent impact of the common monetary policy and of the common fiscal rules. Due to the austerity policies meant to recover competitiveness, real devaluation ensued, with both lower employment rates and substantial wage cuts. Rocketing risk premia increased the spread of the Eurozone's sovereign bonds *vis-à-vis* the German 10-year bund, with particularly high hikes in peripheral countries due to a contagion effect triggered by the partial default of the Greek public debt (Croci Angelini et al. 2016).

The burden of the labor market adjustment has disproportionally fallen on the low-skilled labor force through lower job protection and lower pay, and to a larger extent on non-standard jobs. The widening top–bottom income inequality has been due more to increasing unemployment than to an enlarging distance between the bottom section of the wage distribution and the average wage, although in a few countries the reduction in earnings—larger for high-income than for low-income households—has slightly reduced income inequality (ILO 2015). Cuts in social expenses reduced the public provision of both monetary transfers and in-kind services. Not only were pensions and unemployment benefits reduced, but the degree of coverage, targeting, and generosity shrank too. These developments impinged not only on the earnings dimension, but also on quality of life, as

shown by relevant variations in the main non-monetary well-being dimensions in different countries. Enduring consequences for living conditions were registered in crucial dimensions of well-being, such as health, education, and also social inclusion (Jenkins et al. 2013). In 1985, the Council of Ministers of the European Union defined the "poor" as: "the persons whose resources (material, cultural and social) are so limited as to exclude them from the minimum acceptable way of life in the Member State to which they belong" (EU Council of Ministers 1985). As an effect of the lowering disposable income caused by countercyclical fiscal impulses and more flexible labor markets, the vulnerability of low-skilled workers and the precarious conditions of the labor force with part-time jobs expanded.

Empirical evidence shows that in two-thirds of OECD countries, income inequality has been growing hand in hand with relative poverty; in all of them the risk of poverty, and in some of them also the intensity of poverty, has been soaring (Atkinson et al. 2010). The percentage of materially deprived people ranged from 3% in Luxembourg and 6% in Sweden and the Netherlands up to 45% in Latvia. These distances were much wider than the dispersion of poverty risk, ranging from 10% to 21% (Fusco et al. 2010, pp. 137–38). However, the heterogeneity in living conditions across European countries extends beyond the income-based indicators of poverty and inequality. An Index of Economic Well-being, constructed by incorporating information on consumption, wealth, inequality, and economic insecurity for the OECD countries, shows that the economic crisis more negatively affected well-being in the Peripheral Eurozone, *vis-à-vis* the other countries in the sample.

Low-income people and the poor were also disproportionately affected by the intensification in non-material deprivation. The risk of poverty soared mainly in the sub-group of involuntarily part-time workers (Horemans et al. 2015). While the most dramatic fall in income was seen in the poor in Greece, empirical evidence shows that in many countries low-income groups and the poor were more severely hit during the Great Recession in terms of socio-economic attributes (Whelan and Maître 2012). Between 2008 and 2012 the fall in equivalized household income was very large in Iceland (40%), Greece (30%), and Ireland (20%), and to a lesser extent in the United Kingdom, Spain, and Portugal, where reductions ranged from 13% to 10%. To convey the well-being distance across people concerning material resources, the at-risk-of-poverty indicator has been employed. In 2008, 19 million people were living in severely materially deprived households in Europe; 17 million individuals aged 0–59 were living in jobless households; 49.6 million were living in households at risk of poverty, but were neither jobless nor severely materially deprived; 40 million were living in jobless households and/or materially deprived even if not at risk of poverty; whereas 6.9 million were living in jobless households, at risk of poverty, and severely materially deprived. "The rate for the 12 'new' Member States (NMS12) was 17.3 per cent, a little but not much higher than for EU-15 with a rate of 16.4 per cent. It is certainly not the case that those at risk of poverty on the EU definition are mostly to be found in the New Member States: of the 80+ million at risk of poverty in EU-27, 64 million are to be found in the EU15. In Germany, alone, there are 12½ million; in the United Kingdom 11½ million; in Italy 11 million; and France and Spain together account for a further 17 million. In the largest New Member State, Poland, the number of people at risk of poverty is about 11½ million" (Atkinson and Marlier 2010, p. 106).

The Europe 2020 Agenda, adopted by the European Union (European Commission 2010), pointed at making substantive progress, among other goals, in the promotion of social inclusion. The objectives set in Lisbon in 2000 were neither entirely accomplished by all member states, nor by the EU as a whole (Grimaccia 2021). Also due to the COVID-19 pandemic, the "strategy for a smart, inclusive and sustainable growth" did not deliver its promises. The 2020 target in the area of poverty and social exclusion was defined on the basis of three indicators: (1) the number of people considered 'at-risk-of-poverty' according to the EU definition, where the poverty risk threshold set at 60% of the national household equivalized median income; (2) the number of materially deprived people; and (3) the number of people aged 0–59 living in 'jobless' households, where no member aged 18–59 is working, or where members aged 18–59 have, on average, very limited work attachment.

The target was set to reduce by 25% the number of Europeans living below national poverty lines, by lifting around 20 million people out of poverty. The problem of fulfilling the objectives for social inclusion set in the Europe 2020 Agenda was considerably exacerbated (Atkinson and Marlier 2010).

The Great Recession hit European countries differently by increasing the risk of poverty and inequality and, within those countries, hit individual households in terms of material and non-material deprivations, which include health and education as well as other dimensions relevant for the quality of life—a concept whose content is still debated, but on whose multidimensionality there is no doubt.

Our paper investigates how the real devaluation ensuing the Great Recession affected the multidimensional well-being in European countries through Principal Component Analysis (PCA). The main idea is to summarize in a few major points the differences, if any, encountered by households after the crisis, how their well-being was affected at that time, and whether or not they found it difficult to recover. While there is no lack of studies addressing single issues at the country level, especially from a macroeconomic point of view, not many are based on microdata and explore household behavior in a multidimensional framework by applying PCA.

The paper is organized as follows: Section 2 reviews some empirical literature meant to frame the setting and where the relevance of multidimensional well-being has been addressed; Section 3 discusses the methodological choice of analyzing the impact of the Great Recession on European well-being through PCA, also compared with other methods. The results are presented in Section 4 and discussed in Section 5 where a comparison of our results with previous research is provided. The 2008 crisis impinged differently on the various dimensions of well-being. All in all, our findings very much differentiate depending on the indicators and on the different groups of countries. Section 6 concludes the paper.

## 2. Literature Review

The concern about income distribution and social exclusion in private households in the EU compellingly emerged at the turn of the century. The European Union Statistics on Income and Living Conditions (EU-SILC) dataset was established in the framework of the Open Method of Coordination within the Programme of Community Action meant to encourage cooperation between Member States to counteract social exclusion. It covers European countries, not necessarily members of the EU, and aims at issuing comparable statistics through an integrated design in this area of inquiry. A very wide literature appeared focusing on concepts, measurement, and evaluations of different aspects of inequality. By their very nature, socio-economic phenomena are the joint product of a variety of micro-economic characteristics (e.g., individual material deprivation) and/or macroeconomic conditions (e.g., an underemployment equilibrium with jobless households and/or individuals at high risk of poverty) impinging on the well-being of society at large while interrelations across the most relevant variables are difficult to disentangle.

Following the "capability approach" (Sen 1985), in recent decades the research on well-being has turned towards multidimensionality. Indeed, quality of life is a multifaceted concept (Nussbaum and Sen 1993) to be achieved through a series of functionings, consisting of opportunities in terms of personal capacities. The empirical strand of literature on social indicators has increased enormously in the last three decades, suggesting that, to obtain a comprehensive evaluation of well-being, more dimensions need be added to the standard monetary dimension (Nolan and Whelan 2014).

Divergent per capita GDP across the EU member states, and the dispersion of socio-economic status within the population, are bound to impinge on health conditions (Crimmins et al. 2009). According to EU-SILC data (which refer to self-perceived health status, longstanding illness or disability, unmet medical and dental treatments, and limitations on daily activity), health limitations impaired activity levels between up to

around one-fifth in Cyprus, Poland, Sweden, and the United Kingdom, and over one-third in Estonia, Finland, and Latvia (Hernández-Quevedo et al. 2010).

An analysis conducted by the EUROMOD team shows that the key factor in protecting a household from a drop in income is the presence of more than one single member of the household earning an income (Figari et al. 2010). Two individuals with the same income can have very different living standards if the resources available to each of them differ because of different national provision of public transfers (Fusco et al. 2010). Welfare institutions were crucial in the reduction of the risk of poverty within the European Union, ranging from under 15% to over 60%, with an average value of 38% (Whelan et al. 2014).

While the evolution of jobs, especially for low-income households, is certainly relevant in the cross-country comparison of well-being in Europe before and after the crisis, the effect of the switch towards a more flexible labor market is difficult to assess. During the Great Recession, unemployment in the OECD labor force rose from 6.6% to 8%, with youth unemployment doubling on average, and reaching a peak of 50% in Greece and Spain (OECD 2015). Recent estimates convey the message that a possible positive impact on the employment rate very much depends on the initial degree of rigidity and the mix of institutional reforms (Sologon and O'Donoghue 2014).

As for accommodation, an OECD Statistical Brief reports that housing prices, along with the savings ratio, represent a key driver of the level of household wealth, as the positive correlation between the median net wealth of households and the annual real growth rate of house prices is strong in the long run (Murtin and Mira d'Ercole 2015, p. 4). The relationship between income and wealth is also influenced in Europe by the varying impact across clusters of EU countries of the different forms of housing tenure (Kemeny 2001; Croci Angelini 2015). To compare the standard of living of owner-occupiers and tenants, the method adopted by Eurostat consists in the "imputation" of a rent to owners (having subtracted the actual housing costs). Overall, the adjustment performed by means of the inclusion of imputed rent reduces the degree of income inequality. In particular, the at-risk-of-poverty rate would fall by five percentage points in Ireland and the United Kingdom, four in Estonia and Spain, and more than two in Belgium, Greece, Latvia, and Portugal (Sauli and Törmälehto 2010). Although home-ownership disproportionately affects the well-being of high-income versus low-income households, the impact of house property on a household's financial balances is heterogeneous. On the one hand, the owner-occupier benefits from a higher income, as he gains a hidden rent (corresponding to the saved rent, which would have been paid to a landlord); also, a retired worker living in his own flat, but on the brink of poverty because of a low pension, could use his house to obtain a loan from a bank so as to improve a poor lifestyle. On the other hand, the loss in household equivalized income during a recession is countered in some countries by offering home-ownership as the collateral to borrowing, while in other countries the mortgage associated with home-ownership may worsen already distressed household finances, mainly depending on the income level of households and the national percentage of home-ownership (Sierminska 2012). In some EU countries the indicators for housing conditions were found to be highly correlated with income, while the indicators of material deprivation usually present a stronger relationship with income than with housing conditions, mainly as an effect of financial stress (Nolan and Whelan 2010). Furthermore, the sudden fall in short-term income impacts everyday life, and a declining long-term income counts more for housing conditions and the social environment; similarly, the degree of deprivation is higher for financial distress than for worsening social environment and housing conditions (Fusco et al. 2010).

In the research effort aimed at evaluating multidimensional well-being (MWB), a methodological issue has to be tackled. The dimension-by-dimension approach—a "large and eclectic dashboard" (Stiglitz et al. 2009), or a "portfolio of indicators" (Atkinson et al. 2002)—aims at preserving the information on the interpersonal dispersion of well-being in each dimension. A synthetic indicator may summarize the overall well-being at the cost of ignoring possible interactions across dimensions. On this issue, the empirical evidence

stemming from the EU-SILC database is unclear. On the one hand, the estimate of three indicators—being at risk of poverty, living in a jobless household, and suffering from material deprivation—shows that one-third of the individuals are "disadvantaged" in more than one dimension (Atkinson et al. 2010, p. 127). On the other hand, everywhere there is a low correlation between the income level and the level of deprivation; in particular, in some countries a high level of deprivation is associated with a low level of poverty (Atkinson and Marlier 2010). To compute an index for each dimension avoids two critical issues: the normative evaluation of the weight to be attributed to each dimension, and the assessment of the degree of substitutability among them (Decancq and Lugo 2013).

## 3. Methodology

Multidimensional well-being (MWB) indices seek the impact on well-being stemming from the mutual reinforcement of conditions often characterized by a high degree of complementarity. The variables they rely upon seldom enjoy orthogonality, a characteristic the lack of which hinders many quantitative analyses. The inputs face the problems of identifying the relevant dimensions, find indicators able to describe them, and aggregate the indicators into a single figure meant to aptly describe the multidimensional phenomenon. From a theoretical point of view, several characteristics are needed, a requirement that has been coped with by axiomatic methodologies (Weymark 2006). Empirically, they are often based on surveys, such as the European Quality of Life Survey (EQLS), where questions are posed by Eurofound to thousands of selected individuals. The queries are obviously designed to fit the purpose of the survey, yet independent researchers may use the data for their own investigations.

To compare well-being in European countries before and after the Great Recession we use data from the EU-SILC dataset for the years 2007 and 2012. The data—covering 26 countries, among which 24 belong to the European Union and two are non-EU countries (Norway and Iceland)—are complete, i.e., all relevant variables exist for both years and all countries. Although Bulgaria, Croatia, Malta, and Romania are EU members today, they have been excluded for incomplete availability of data. Our units of analysis are these 26 European countries: for each of them the dataset includes several thousand entries, based on both households and individuals, from which each country's information is calculated.

The Principal Component Analysis (PCA) searches for the unknown factors which are at the roots of the well-being outcomes. This methodology consists of the computation of mutually orthogonal principal components (through the linear combinations of the original variables, i.e., the different indicators considered for each dimension). A large number of initial, possibly correlated, indicators are transformed into mutually uncorrelated linear combinations. The principal components are extracted with the aim of identifying hidden, unobservable variables able to explain a major portion of the variance. Hence, dispersed information about each individual entry is concentrated in principal components, each one summarizing the information conveyed by a larger set of indicators. The construction of composite indices from individual indicators helps in comparing across time and space the performance of a unit based on a large amount of information (Freudenberg 2003). The PCA is a reliable method meant to overcome the trade-off between comprehensiveness (which compresses the variety of dimensions of life into a synthetic index) and meaning (whereby the focus on the impact of the crisis on well-being prompts preserving the distinct short-term evolutionary path of well-being in each dimension) and so helps in weighting performances and devising policies (Nardo et al. 2008). To evaluate unobservable variables such as well-being or quality of life, an alternative method is the fuzzy set approach (Betti 2016) where the methodological focus is on the appropriate weights, while another method is by axiomatic measurement, which keeps a desirable decomposition characteristic and was proposed in a previous paper (Croci Angelini and Michelangeli 2012).

Our methodological choice in favor of PCA was based on the reduction of variables aimed at understanding the structure underlying a large list of interrelated indicators in order to reduce them to a small number of aggregate synthetic variables. The search

for latent variables is particularly suitable for dealing with the large amounts of data characterizing a socio-economic survey on income, quality of life, and living conditions such as those provided by EU-SILC, the dataset we rely upon.

By sorting 68 variables from the EQLS dataset, Betti (2016) identifies eight relevant groups, ranging from quality of relations to subjective well-being, and including health and housing quality. On the contrary, we consider the five dimensions we regard as most relevant for our research. With the three most investigated variables in the analysis of multidimensional well-being (income, education, and health), this paper addresses employment and accommodation. These two additional dimensions are relevant for the lack of territorial homogeneity in Europe, where across-country mobility is more demanding and risky, due to institutional as well as cultural differences.

EU-SILC includes both subjective and objective queries. While we aim to keep as much information as possible from the dataset, our strategy is to rely upon objective information only and to check its coherence with subjective evaluations, when available. This check is important especially when the eligible answers are not dichotomous and allow either a scale of preferences, or provide reasons to support the answer. A case in point is when the answer "Yes" differentiates between intensities, or the answer "No" distinguishes between preference (those who do not want the item) and feasibility (those who cannot afford it).

Following the OECD, both material and non-material sources of well-being have been considered. However, our classification does not exactly reflect the same sub-sets of dimensions separating the more observable material living conditions from the (somewhat more) subjective quality of life. The OECD framework for the assessment of well-being considers outcomes and their distribution across the population achieved in two broad domains: material living conditions (i.e., income and wealth, jobs and earnings, and housing conditions) and quality of life (i.e., health status, work-life balance, education and skills, social connections, civic engagement and governance. environmental quality, personal security, and subjective well-being).

As for the material sources of well-being, monetary and non-monetary dimensions have been chosen, relevant at the household level and quantified through three observable indicators:

1. Monetary income, which singles out individuals with an equivalized disposable income below 60% of their national median and is assessed through the dichotomous variable "at-risk-of-poverty". Equivalized disposable income corresponds to the total household income after social transfers, available for spending or saving, divided by the number of household members, converted into adults, according to the OECD modified equivalence scale.

2. Material deprivation, which refers to difficulties in everyday life stemming from lack of key provisions. The Eurostat definition covers a set of sub-indicators of economic strain, related to a household's inability to cope with a number of items deemed essential for a decent life. According to this definition, materially deprived individuals are those who cannot afford at least three out of the following list of nine items: (1) paying for unexpected financial expenses, (2) keeping their homes adequately warm, (3) eating meat, fish, or a protein equivalent every second day, (4) enjoying a week's holiday away from home, (5) having a car, (6) having a washing machine, (7) having a color TV, or (8) having a telephone, and (9) paying mortgage, rent, utility bills, as well as hire purchase installments or other loan payments without delay.

3. Housing deprivation, which looks at poorly-comfortable dwellings. Following the Eurostat definition, to be severely deprived, a household should live in an overcrowded dwelling, which also suffers from at least one of the following deficiencies: (1) being too dark, (2) having a leaking roof, damp walls/floors/foundation, or rot in the window frames or floor, and (3) having neither a bath, nor a shower, nor an indoor toilet. The Eurostat methodology for the calculation of the overcrowding variable employs auxiliary variables referring to the number, age, and gender of the household members.

As for non-material sources of well-being, three dimensions, relevant at the personal level, have been considered and measured by indicators as objective as possible:

1. Health, which collects all objective information available in EU-SILC on this dimension. The variables included are: (1) suffering from a chronic illness, (2) unmet medical treatment, (3) unmet dental treatment, and (4) activities limited by impaired health conditions. Information about personal general health assessments was deemed subjective and was overlooked.

2. Education, which measures young individuals not in employment, education, or training (NEETs)—a condition which points at the mismatch between jobs and education. This variable was computed by selecting the individuals aged 16–29 whose economic activity was non-existent and who participated neither in education nor in training.

3. Economic activity, which is summarized by three indicators related to difficulties suffered in the labor market. The variables included are: (1) temporary jobs, (2) unemployment, and (3) under-employment, appraised by work intensity, which refers to households where, during the previous 12 months, their components aged 18–59 worked less than 20% of their total potential.

## 4. Results

Table 1 shows the correlation matrix of these indicators for the year 2007. The highest correlations are between both deprivations (0.859) and between both unmet treatments (0.824); correlations over 0.5 are also observed between chronic illness and limited activities; for severe housing deprivation and NEETs as well as unmet treatments, NEETs are also associated with unemployment and underemployment. A weak negative correlation is observed between chronic illness and poverty risk (−0.061) and, in turn, the NEETs (−0.138), as well as between unmet dental treatment and unemployment (−0.229), while temporary jobs show limited negative correlations with housing deprivation and all items connected to health.

**Table 1.** Inter-item correlation matrix for the year 2007.

|  | 1. | 2. | 3. | 4. | 5. | 6. | 7. | 8. | 9. | 10. | 11. |
|---|---|---|---|---|---|---|---|---|---|---|---|
| 1. Severe housing deprivation | 1 | | | | | | | | | | |
| 2. Suffer from chronic illness | 0.194 | 1 | | | | | | | | | |
| 3. Unmet medical treatment | **0.663** | 0.194 | 1 | | | | | | | | |
| 4. Unmet dental treatment | **0.510** | 0.087 | **0.824** | 1 | | | | | | | |
| 5. Activity limited by bad health | 0.396 | **0.591** | 0.352 | 0.175 | 1 | | | | | | |
| 6. Poverty risk | 0.385 | −0.061 | 0.383 | 0.381 | 0.220 | 1 | | | | | |
| 7. Extreme material deprivation | **0.859** | 0.093 | **0.714** | **0.605** | 0.347 | 0.415 | 1 | | | | |
| 8. Unemployment | 0.366 | 0.140 | 0.046 | −0.229 | 0.398 | 0.161 | 0.337 | 1 | | | |
| 9. Underemployment | 0.311 | 0.033 | 0.396 | 0.245 | 0.240 | 0.267 | **0.507** | 0.415 | 1 | | |
| 10. Temporary jobs | −0.043 | −0.200 | −0.057 | −0.067 | −0.107 | 0.140 | 0.046 | 0.451 | 0.490 | 1 | |
| 11. NEETs | **0.509** | −0.138 | 0.300 | 0.113 | 0.242 | 0.427 | **0.545** | **0.685** | 0.481 | 0.364 | 1 |

Source: Own calculations on EU-SILC 2007 and EU-SILC 2012 dataset. Correlations over 0.5 in bold.

Table 2 shows the correlation matrix of the same indicators for the year 2012. Again, the highest correlations are observed between both deprivations (0.864) and between both unmet treatments (0.705). However, more impressive variations stand out. With respect to 2007, the unemployment correlation increases with poverty risk (from 0.161 to 0.69) and between unemployment and underemployment (from 0.415 to 0.75). NEETs are no longer correlated with any other variable and neither are Temporary jobs. The number of (weak) negative correlation values has also increased, mainly affecting those suffering chronic illness and those employed in temporary jobs.

**Table 2.** Inter-item correlation matrix for the year 2012.

|  | 1. | 2. | 3. | 4. | 5. | 6. | 7. | 8. | 9. | 10. | 11. |
|---|---|---|---|---|---|---|---|---|---|---|---|
| 1. Severe housing deprivation | 1 | | | | | | | | | | |
| 2. Suffer from chronic illness | 0.061 | 1 | | | | | | | | | |
| 3. Unmet medical treatment | **0.583** | 0.190 | 1 | | | | | | | | |
| 4. Unmet dental treatment | 0.349 | 0.094 | **0.705** | 1 | | | | | | | |
| 5. Activity limited by bad health | 0.217 | **0.545** | 0.008 | −0.233 | 1 | | | | | | |
| 6. Poverty risk | 0.273 | −0.227 | 0.325 | 0.294 | −0.003 | 1 | | | | | |
| 7. Extreme material deprivation | **0.864** | −0.072 | **0.515** | 0.353 | 0.088 | 0.448 | 1 | | | | |
| 8. Unemployment | 0.325 | −0.221 | 0.241 | 0.375 | −0.018 | **0.690** | 0.487 | 1 | | | |
| 9. Underemployment | 0.328 | −0.162 | 0.382 | **0.616** | −0.170 | **0.567** | 0.447 | **0.750** | 1 | | |
| 10. Temporary jobs | −0.066 | −0.125 | 0.138 | 0.167 | −0.125 | 0.299 | −0.114 | 0.479 | 0.327 | 1 | |
| 11. NEETs | 0.186 | 0.085 | 0.184 | 0.105 | −0.014 | 0.245 | 0.370 | 0.141 | 0.191 | −0.046 | 1 |

Source: Own calculations on EU-SILC 2007 and EU-SILC 2012 dataset. Correlations over 0.5 in bold.

A factor analysis—a method of checking the dimensionality of the scale on the basis of the internal consistency of all items—was performed following the Cronbach's alpha test on reliability. The aim of this exploratory exercise is to single out groups of related variables, each one describing some part of the dimensions we are interested in, but unfit to take part in a regression model. The principal components are extracted from the groups of variables described above.

The results of these tests, performed on the dataset to identify reliability, sampling adequacy, and data suitability, are shown below in Table 3 for both years before and after the outbreak of the Great Recession.

**Table 3.** Tests on reliability, sampling adequacy, and data stability.

| **Test on Reliability** | | **2007** | **2012** |
|---|---|---|---|
| Cronbach's alpha | | 0.757 | 0.649 |
| Cronbach's alpha based on standardized items | | 0.820 | 0.763 |
| Item number | | 11 | 11 |
| **Test on Sampling Adequacy** | | **2007** | **2012** |
| Measure of sampling adequacy KMO (Keiser Meyer Olkin) | | 0.709 | 0.634 |
| **Test on Data Suitability** | | **2007** | **2012** |
| Bartlett's Test of sphericity | Approximate Chi-square | 153.240 | 139.678 |
| | Degrees of freedom | 55 | 55 |
| | Significance | 0.000 | 0.000 |

Source: Own calculations on EU-SILC 2007 and EU-SILC 2012 dataset.

The PCA is a reduction technique able to summarize a large set of variables into a smaller number of components so as to seek a pattern resulting from the correlation existing among them. All indicators enter the PCA after being transformed into dichotomous variables. Therefore, information about the causes of difficulties and/or the degree of deprivation when existing at this stage is lost.

The extraction of the components has been performed by employing two methods: the Kaiser eigenvalues (reflecting the components' variance) and the Scree plot test (plotting the eigenvalues and evaluating the shape of the line) so as to seek a good balance between efficiency (i.e., keep the number of components as low as possible so as to describe the results as simply as possible) and completeness (i.e., not to drop any relevant information).

The varimax method, by the rotation matrix of the components, has been applied in order to minimize the number of indicators and keep as much of their information content as possible in the analysis in a situation where many variables were supposed to be cross-correlated (or this possibility could not be ruled out).

Table 4 shows the explanatory power of the three components for the year 2007—and four components for the year 2012—ranked according their decreasing importance. All

Kaiser eigenvalues are higher than 1.0, so that in principle all components are worth being considered in our analysis as the variance they can explain is higher than that explained by the single variables.

**Table 4.** Total variance explained—rotated factor loadings.

| | Total | | % Variance | | % Cumulative | |
|---|---|---|---|---|---|---|
| **Component** | **2007** | **2012** | **2007** | **2012** | **2007** | **2012** |
| 1 | 3.487 | 2.533 | 31.700 | 23.027 | 31.700 | 23.027 |
| 2 | 2.674 | 2.255 | 24.306 | 20.498 | 56.006 | 43.525 |
| 3 | 1.857 | 2.048 | 16.882 | 18.622 | 72.888 | 62.147 |
| 4 | - | 1.597 | - | 15.520 | - | 76.668 |

Source: Own calculations on EU-SILC 2007 and EU-SILC 2012 dataset.

In 2007, nearly 73% of the total variance was explained by synthetizing all our information into three components, among which the first one alone contributes to almost one-third of it all. In 2012, a fourth component needed to be extracted to reach over 76% of the total variance explained, while none of them could yield such a high explanatory power. This could be a hint that the crisis has deeply changed the social environment.

Table A1 in Appendix A shows the structure of the components extracted by the PCA. In the year 2007, the first year of our inquiry, i.e., before the Great Recession, the 11 indicators, in turn yielding evidence for our dimensions of well-being, were reduced into the three components, whose explanatory power was presented in Table 4. The first component contains five indicators (items) related to miserable conditions: 1. Severe housing deprivation, 2. Unmet medical treatment, 3. Unmet dental treatment, 4. Poverty risk, and 5. Extreme material deprivation. The second component contains four indicators all related to the worsening of labor market conditions: 1. Unemployment, 2. Under-employment, 3. Temporary jobs, and 4. NEETs. The third component comprises two indicators regarding health conditions: 1. Suffering from chronic illness, and 2. Activities limited by bad health.

In the year 2012, the Great Recession struck the well-being conditions in many countries and the picture changes accordingly. The first component mainly contains indicators signaling difficulties in the labor market (unemployment, under-employment, and temporary jobs) adding to poverty risk (the only indicator present in the former first component), for a total of four indicators. The second component collects three variables only: NEETs, severe housing deprivation, and extreme material deprivation. This reshuffling of items between the first and second component suggests that the consequences of the financial crisis spread over the weak functioning of the labor market in a multidimensional manner ranging from housing and material deprivation to the NEETs phenomenon. The third component contains two indicators previously included in the first component: unmet medical and dental treatments, while the fourth component, needed to reach a similar level of explained total variance as in 2007, again refers to health problems and is composed by the same indicators included in the third component in 2007. This shows that the impact of the global financial crisis took only a few years to cause a very profound and asymmetric change. How each country contributed to the components is shown in Table A2 in Appendix A, with a graphical representation of the impact of the first and second component (Figure A1).

A cluster analysis delivers a clearer understanding of the PCA results. A Hierarchical Clustering on Principal Components was executed according to Ward's method with Euclidean distance matrix and Z-scores.

The dendrogram plots, clustering the 26 European countries in 2007 and in 2012, are shown in Figure 1a and Figure 1b, respectively. Various groups and subgroups of countries can be identified according to the chosen level of aggregation: the lower the level, the higher the number of sub-groups of countries.

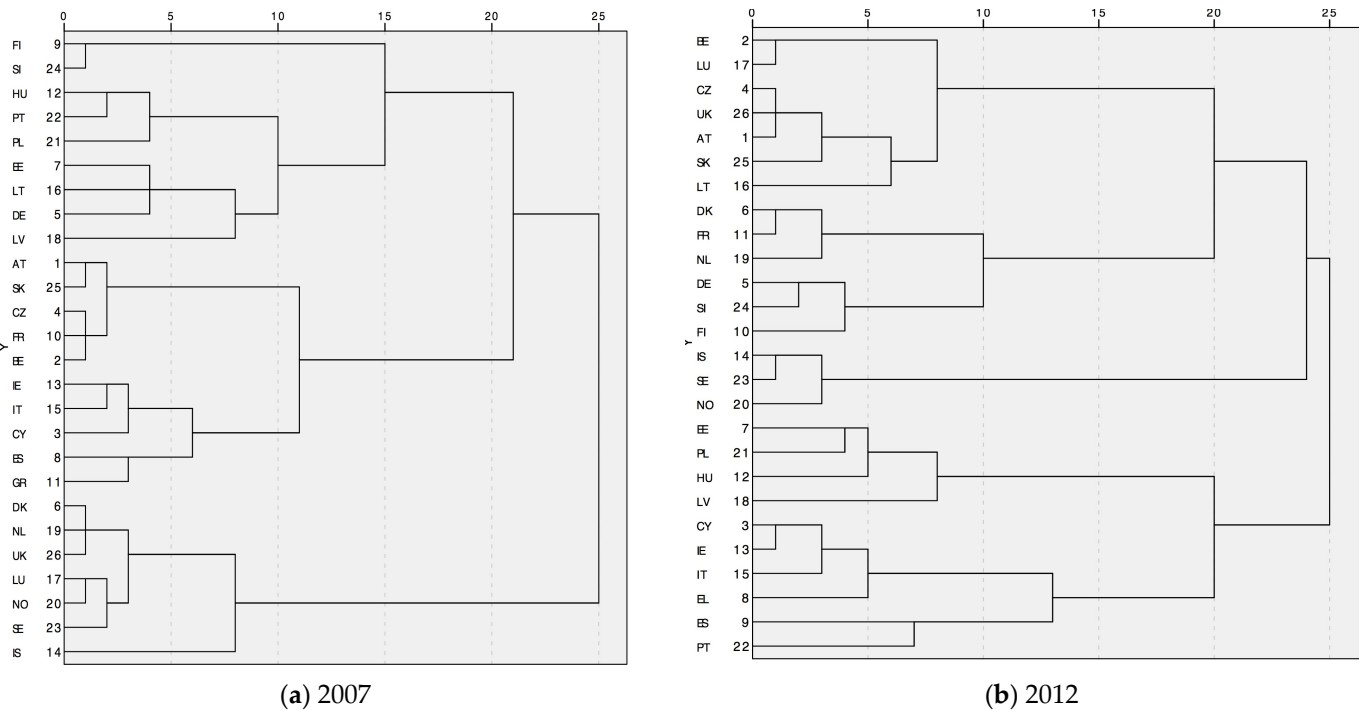

**Figure 1.** Dendrogram clustering the 26 countries in 2007 (**a**) and 2012 (**b**).

In 2007, the group composed by Denmark, the Netherlands, the United Kingdom, Luxembourg, Norway, Sweden, and Iceland (i.e., mainly Scandinavian and rich countries) stands alone even below the level of aggregation 10. The second main group can be subdivided into two subgroups containing: (1) Germany and many north-eastern countries (Finland, Slovenia, Hungary, Portugal, Poland, Estonia, Lithuania, and Latvia) and (2) France and many central and southern countries (Austria, Slovakia, Czechia, Belgium, Ireland, Italy, Cyprus, Spain, and Greece).

In 2012, a deep reshuffling also occurred among the clusters, so that even the composition of the former two main groups changed. All dimensions, not just the various sources of household material resources, contribute to a differing extent in magnifying the heterogeneity of well-being across Europe.

At a comparable level of aggregation in Figure 1b five subgroups can be identified from the top to the bottom: (1) Belgium, Luxembourg, Czechia, the United Kingdom, Austria, Slovakia, and Lithuania, (2) Denmark, France, the Netherlands, Germany, Slovenia and Finland, (3) Iceland, Sweden, and Norway, all belonging to the first group, (4) Estonia, Poland, Hungary, and Latvia, and (5) Cyprus, Ireland, Italy, Greece, Spain and, Portugal, belonging to the second group.

While the dendrograms gather the 26 countries according to the hierarchical clustering on all principal components, Table A2 in Appendix A provides a breakdown of how each country contributes to the PCA results. The interested reader may check how individual countries fare and compare before and after the Great Recession.

## 5. Discussion

In the appraisal of our results, one should keep in mind that, going from 2007 to 2012, the item content of the components has changed. In 2007, the first component mainly shows people's conditions for risk of poverty, material deprivation in terms of private goods for a decent everyday life, and the quality of housing conditions as well as some health issues. The second component indicates shortcomings in economic activity related to the labor market. One could think of poor households, which have deprivation as the first problem in their well-being assessment and for whom changes in the labor market and organization would perhaps be able to alleviate their deprived circumstances. In 2012 these issues,

except the NEETs, are included in the first component together with poverty risk, while the second component includes the NEETs with housing and material deprivations. The priority seems to have changed in the consideration of the respondents well-being. It seems that they feel being at risk of poverty for the threat of losing their job. The Great Recession seems to have changed the priorities in the ranking of the difficulties encountered by European households. The contours of disadvantage in 2007, before the crisis, mainly refer to deprivation, which might be caused by difficulties on the labor market, while in 2012 the main weakness is found in troubles in the labor market, leading to poverty. This change does not seem to be short-lived. The analysis of social insecurities by Eurofound (2018), which is based on data gathered in the European Quality of Life Survey (EQLS) across the 28 EU Member States, underlines that, due to the consequences of the 2008 financial crisis, many more people feel insecure about their future. The exposure to vulnerability has reached people who have suddenly become at risk of job loss, or are loaded with over-indebtedness, or are unable to pay for healthcare. Similar results are shown by Ayllón and Gábos (2017) who employ EU-SILC and observe that the three indicators put forward in the Europe 2020 Agenda—being at risk of poverty, severe material deprivation, and low work intensity—present state dependence: in the majority of European countries, once one household is caught by these economic hardships, the persistence is such that is very difficult to escape those conditions, and of course this adds to insecurity. In 2017 at the informal summit held in Gothenburg, the European Pillar of social rights was proclaimed, also envisaging a multidimensional indicator named "at risk of poverty or social exclusion" (AROPE), which includes the sum of people who are either at risk of poverty, or severely materially and socially deprived, or living in a household with a very low work intensity (Atkinson et al. 2017). Some issues characterizing the multidimensional nature of the socio-economic environment have been collected in a single multidimensional indicator summarizing the EU2020 target. Unfortunately the statistics about the single variables have been discontinued, which makes it difficult to extend a full comparison to nowadays.

One of our main findings is that in 2012 employment-related issues stand out as a major determinant of the worsening of the quality of life. This result can be traced back to the negative impact of the demise of EPL of the hoarding of low-skilled workers during the crisis, as argued in Fedotenkov et al. (2024) who employ microdata and specifically address the effects of the Great Recession on labor productivity, detailed by country and sector.

The discussion continues by looking at the different groups of countries.

In trying to make a complex matter simpler, we will keep identifying the groups on their geo-economic position, i.e., Southern, Eastern, Northern, and Western. Exceptions may be checked by looking at the single country's results.

The financial crisis was counteracted with greater success by the richer Western continental countries. The socio-economic performance of the Netherlands, Denmark, Luxembourg, Germany, and Austria was fairly satisfying in 2007 and it looks as if these countries totally recovered by 2012. France and Belgium even more than recovered. The Nordic countries (Iceland, Norway, and Sweden) more than recovered, too.

Eastern European countries had very much benefitted from the economic integration with the German productive system after the demise of the Soviet Union and, to a larger extent, after the 2004 adhesion to the European Union. The recession particularly hit these countries mainly because of a very distressed labor market exacerbating socio-economic conditions; however, deprived people do not show problems in the labor market (Croci Angelini et al. 2020).

In 2007, before the crisis, Hungary and Poland were found in the "bad" quadrant (see Figure A1), the one with difficulties in both first and second components; Slovenia, Slovakia and Czechia had failures in the labor market (second component), while the Baltic countries (Estonia, Latvia, and Lithuania) had no major problems with deprivations (first component).

In 2012, Poland slightly improves, but still remains in the "bad" quadrant, while Hungary joins the Baltic countries. It seems that Slovenia, Slovakia, and Czechia have

reacted to the crisis by struggling to keep the labor market afloat while reducing both deprivations and so reach a position near to zero at the crossing of the two components.

The poor performance of Southern European countries stands out, as the crisis struck them worse of all. Italy and Greece are in the "bad" quadrant in 2007 and are still there in 2012, joined by Cyprus, and aggravate their conditions further away from the zero. Spain and Portugal show improvements in the labor market at the expense of more deprivations caused by deregulation.

In these countries the crisis is far from over also in the following years. Although it has not been possible to compute a recent comparable PCA for the lack of some variables, Southern countries need at least three more years (Portugal) to recover material deprivation levels, while Greece so far has not yet recovered. As for the labor market-related component, unemployment rates in Spain and in Greece were still over 10% in 2018. No other country shows the same record.

Our results are also largely coherent with the findings of Ivanová et al. (2022) who explore EU quality of life through 19 EU-SILC based variables and employ PCA to reduced them to five factors (material-economic conditions, social contacts and existential issues, environmental issues and quality of environment, health limitations, and crime), which are the most important factors affecting EU inhabitants' quality of life. Their study shows the maximum positive correlation (0.93) between households making ends meet with great difficulty and arrears (mortgage or rent, utility bills, or hire purchase); also, the quality of life has been found to be substantially negatively influenced by social insecurity mainly as an effect of economic safety.

The research work by Mazurek (2016) carried out on quarterly macrodata and interested in the magnitude and shape of the Great Recession on 25 EU countries confirms our findings that the periphery of Europe (i.e., mainly East and South) has been the area mostly hit by the Great Recession.

Finally, as for the remaining health-related components, it is perhaps worth mentioning that the performance of none of the Southern countries was in danger in 2007 (together with the Nordic and some other countries), while it was in Portugal in 2012.

All in all, our analysis suggests that the countries with both most valuable economic structure and socio-economic indicators—again the Western continental and the Nordic countries—have shown a remarkable resilience during the period following the financial crisis, confirming their performance both for component 1 and 2. A significant exception is the UK, which is singled out for the worsening of component 2.

Indeed, as shown in Betti (2016) the most relevant change happens in each single group for quality of life rather than in the overall index.

## 6. Conclusions

The above discussion on the evolution of well-being in Europe indicates that the Great Recession has consolidated the division across the four groups of countries. On the one hand we see that the Central-Western and Nordic countries were able to react to the worsening socio-economic conditions; on the other hand the Central-Eastern countries' convergence has stopped and the Southern countries have further been left behind. The UK is a case in point as its performance has worsened after Brexit in 2016. In other words, the crisis has negatively impinged on the capacity of national and supranational institutions in sustaining the market integration process. The evolution of the well-being conditions has been limited to a strengthening of market integration within the four groups of countries. Due to the impact of the crisis the objective of socio-economic convergence has been set aside.

The purpose of this paper was to evaluate variations in aggregate economic welfare within 26 European countries, by connecting the impact on the main dimensions of well-being of the Great Recession, which was very heterogeneous across Core, Peripheral, and Central-Eastern Europe in particular at the bottom of the income distribution.

In some dimensions, such as risk of poverty, unemployment, and material deprivation, where a recession typically provokes negative effects in the short run, evidence shows that the crisis has worsened the well-being of a sizable number of households, mainly in Southern and Central-Eastern Europe. In some other dimensions, such as health conditions, educational achievements, and housing, the impact of the crisis is not particularly relevant. In the European Union, these indicators regard mainly publicly provided services, which are less subject to the decay that has hit automatic stabilizers after the negative fiscal impulses imposed by Brussels to repair distressed public finances. Typically, the possibly impact of the Great Recession on these dimensions could be assessed only in the medium term, in case the size of the cut to public expenditures would be so large as to gravely worsen the provision of these essential merit goods.

The results of our paper can be compared with those reached by studies aimed at the evaluation of the effects of the Great Recession on European countries (e.g., Mazurek 2016; Fedotenkov et al. 2024) and of the countries' performance towards the EU2020 targets (e.g., Grimaccia 2021). Overall, our results are compatible with them, although the aims, scope, and methods may differ.

As for the method, PCA was applied by Ivanová et al. (2022) over 19 variables to assess the quality of life in member countries of the European Union in eight dimensions reduced to the five most important factors. The advantage of this method lies in the way that a wide amount of information is summarized in few major components which may be more important for policy advice rather than a single figure that is only able to rank the countries.

**Author Contributions:** Conceptualization, E.C.A., F.F. and S.S.; methodology, E.C.A. and S.S.; software, S.S.; validation, E.C.A. and S.S.; formal analysis, S.S.; investigation, S.S.; resources, S.S.; data curation, E.C.A.; writing—original draft preparation, F.F.; writing—review and editing, F.F.; visualization, S.S.; supervision, E.C.A.; project administration, E.C.A.; funding acquisition, E.C.A. and S.S. All authors have read and agreed to the published version of the manuscript.

**Funding:** This research received no external funding.

**Data Availability Statement:** All EU-SILC data employed in this paper are available at https://ec.europa.eu/eurostat/web/income-and-living-conditions/information-data (accessed on 6 May 2024).

**Acknowledgments:** While taking joint responsibility for this paper the authors gratefully acknowledge the support, discussion, and valuable comments by Fabio Clementi, Cristina Davino, and Enzo Valentini. We also thank the anonymous referees.

**Conflicts of Interest:** The authors declare no conflicts of interest.

### Appendix A

In 2007, the 11 indicators listed in the first column of Table A1 were reduced into the three components whose explanatory power is presented in Table 4.

**Table A1.** Rotated component matrices [a] for the years 2007 [b] and 2012 [c].

| | 1st Component | | 2nd Component | | 3rd Component | | 4th Component | |
|---|---|---|---|---|---|---|---|---|
| | **2007** | **2012** | **2007** | **2012** | **2007** | **2012** | **2007** | **2012** |
| 1. Severe housing deprivation | **0.750** | 0.095 | 0.269 | **0.772** | 0.324 | 0.376 | - | 0.173 |
| 2. Suffer from chronic illness | 0.021 | −0.224 | −0.144 | −0.102 | **0.870** | 0.264 | - | **0.835** |
| 3. Unmet medical treatment | **0.900** | 0.126 | 0.011 | 0.320 | 0.177 | **0.826** | - | 0.133 |
| 4. Unmet dental treatment | **0.896** | 0.241 | −0.196 | 0.087 | −0.033 | **0.899** | - | −0.101 |
| 5. Activities limited by bad health | 0.232 | 0.028 | 0.186 | 0.181 | **0.834** | −0.231 | - | **0.895** |

**Table A1.** *Cont.*

|  | 1st Component | | 2nd Component | | 3rd Component | | 4th Component | |
|---|---|---|---|---|---|---|---|---|
|  | **2007** | **2012** | **2007** | **2012** | **2007** | **2012** | **2007** | **2012** |
| 6. Poverty risk | **0.555** | **0.749** | 0.291 | 0.376 | −0.076 | 0.055 | - | −0.076 |
| 7. Extreme material deprivation | **0.825** | 0.208 | 0.334 | **0.891** | 0.190 | 0.275 | - | −0.001 |
| 8. Unemployment | −0.051 | **0.883** | **0.855** | 0.269 | 0.368 | 0.122 | - | −0.062 |
| 9. Underemployment | 0.372 | **0.691** | **0.642** | 0.236 | 0.027 | 0.428 | - | −0.177 |
| 10. Temporary jobs | −0.092 | **0.730** | **0.735** | −0.395 | −0.297 | 0.126 | - | 0.004 |
| 11. NEETs | 0.344 | 0.071 | **0.792** | **0.535** | 0.034 | −0.001 | - | −0.014 |

Source: Own calculations on EU-SILC 2007 and EU-SILC 2012 dataset. [a] Numbers in bold identify the items composing the different components after the convergence by Varimax method of rotation and Kaiser normalization. [b] rotation reached convergence criteria in 4 iterations. [c] Rotation reached convergence criteria in six iterations.

Table A2 shows how the 26 countries contribute to the three (2007) or four (2012) components.

**Table A2.** Countries' contributions to the principal components.

| | Country | 2007 1st Component | 2012 1st Component | 2007 2nd Component | 2012 2nd Component | 2007 3rd Component | 2012 3rd Component | 2012 4th Component |
|---|---|---|---|---|---|---|---|---|
| AT | Austria | −0.8119 | −0.7708 | −0.05874 | 0.2812 | 0.04249 | −1.20.05 | 0.11156 |
| BE | Belgium | −0.91417 | −0.26396 | 0.52803 | −0.28846 | −0.37991 | −0.87335 | −1.05017 |
| CY | Cyprus | 0.53777 | 0.41449 | −0.37921 | 0.19816 | −1.00462 | 0.18321 | −0.74536 |
| CZ | Czechia | −0.83171 | −0.86943 | 0.69367 | 0.09863 | −0.01061 | −0.52352 | −0.43346 |
| DE | Germany | −0.2344 | −0.05314 | −0.20694 | −0.65822 | 1.24639 | −0.74793 | 1.45954 |
| DK | Denmark | −0.51385 | −0.6477 | −1.49942 | −0.37235 | −0.30065 | −0.14205 | 0.08512 |
| EE | Estonia | 1.07845 | −0.47987 | −1.03448 | 0.7549 | 1.5694 | 0.32549 | 1.36469 |
| EL | Greece | 0.52409 | 1.98103 | 1.6437 | 1.08254 | −1.72526 | −0.04791 | −0.82843 |
| ES | Spain | −0.60796 | 3.14244 | 1.63703 | −1.26.12 | −1.08996 | −0.27919 | −0.43654 |
| FI | Finland | −1.24456 | 0.04651 | 0.38496 | −0.90573 | 1.95109 | 0.02104 | 2.58969 |
| FR | France | −0.5454 | −0.07848 | 0.75137 | −0.78641 | 0.07443 | 0.39423 | 0.25036 |
| HU | Hungary | 0.79412 | −0.45669 | 0.85781 | 2.01621 | 0.91443 | 0.58656 | 0.18496 |
| IE | Ireland | −0.47203 | 0.5942 | 0.40587 | −0.0914 | −0.94683 | −0.33155 | −1.40245 |
| IS | Iceland | 0.15807 | −1.18332 | −1.64814 | −1.17504 | −2.09799 | 1.47602 | −0.74238 |
| IT | Italy | 0.46134 | 0.77587 | 0.72078 | 0.86241 | −0.92076 | −0.14621 | −0.3447 |
| LT | Lithuania | 1.21276 | −0.0457 | −0.29146 | 1.82744 | 0.36067 | −1.18555 | −0.40005 |
| LU | Luxemburg | −0.6931 | −0.5774 | −0.91574 | −0.45614 | −0.45531 | −0.83776 | −1.5566 |
| LV | Latvia | 3.31357 | −0.2101 | −0.41543 | 1.97079 | 0.7569 | 2.7584 | 0.5087 |
| NL | Netherlands | −0.64051 | −0.82843 | −1.51746 | −0.9206 | 0.0243 | −0.78992 | 0.82626 |
| NO | Norway | −0.2299 | −1.36951 | −0.96034 | −1.1438 | −0.54461 | 0.52326 | −1.20646 |
| PL | Poland | 1.38332 | 0.36321 | 1.47278 | 0.36934 | −0.0028 | 1.02713 | 0.33069 |
| PT | Portugal | 0.16762 | 1.45223 | 1.26733 | −0.96893 | 0.53069 | 1.33337 | 0.26141 |
| SE | Sweden | 0.22784 | −0.53582 | −0.88212 | −1.39502 | −0.0403 | 1.49465 | −0.41712 |
| SI | Slovenia | −1.31893 | 0.4255 | 0.29688 | −0.11817 | 1.7861 | −1.32586 | 1.71076 |
| SK | Slovakia | −0.47952 | 0.03088 | 0.34155 | 0.39451 | 0.30933 | −0.70355 | 0.4685 |
| UK | United Kingdom | −0.32102 | −0.85603 | −1.19227 | 0.68525 | −0.04663 | −0.98394 | −0.58852 |

Source: Own calculations on EU-SILC 2007 and EU-SILC 2012 dataset.

In Figure A1 the first and second component of both years are plotted in the scatter diagrams 2007 (a) and 2012 (b) into four quadrants. Countries that have the heaviest problems are represented in the upper right quadrant, while countries that fare better are in the double negative lower left quadrant. The two diagrams compare the two years with a warning: in 2007, the first and second components together explain more than half the variance (56%), while in 2012 they only reach 43,5%. A three-dimensional diagram for 2012 would reach 62% (and nearly 73% in 2007) but would not be as reader-friendly.

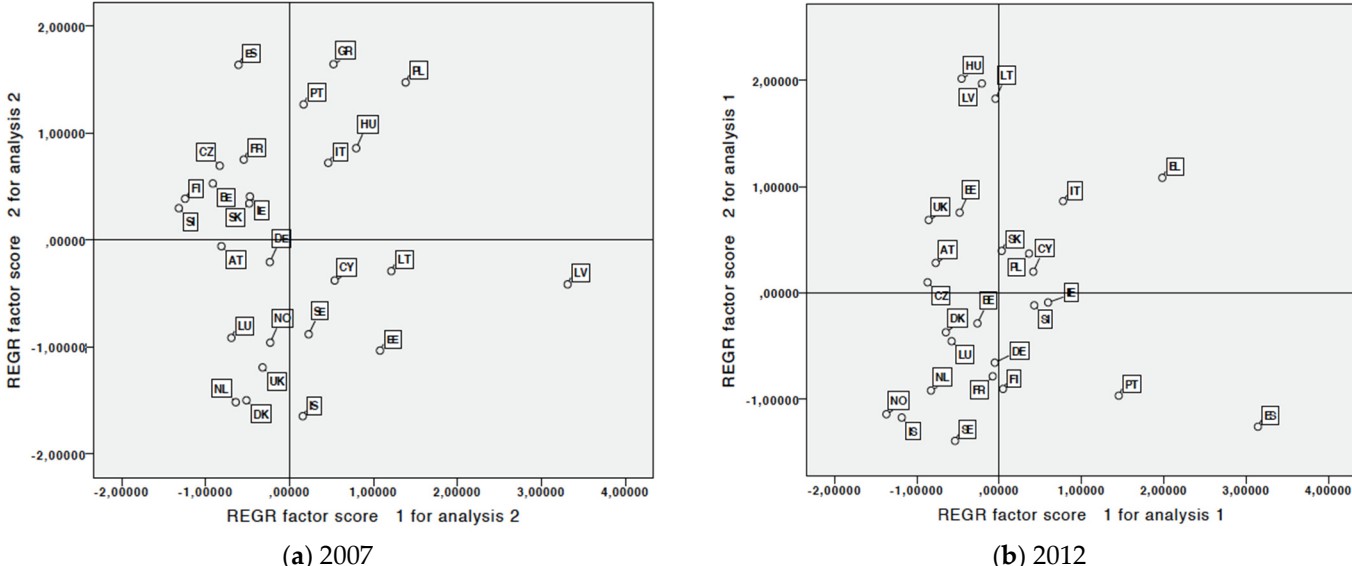

**Figure A1.** Scatterplot of the first and second component for the 26 countries, year 2007 (**a**) and 2012 (**b**).

The first component is measured on the horizontal axis and the second component on the vertical axis. Countries represented on the left quadrants show negative values for the first components (i.e., they tend not to be deprived in 2007, nor have major problems on the labor market in 2012), while countries in the lower quadrants show negative values for the second component. Therefore, countries having both negative values tend to fare better than all the rest, while countries having both positive values tend to be more miserable than the remaining countries.

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
