# Peer review of "The Impact of the Great Recession on Well-Being across Europe Ten Years On: A Cluster Analysis"

_economies, doi:10.3390/economies12050115_

Round 1

Reviewer 1 Report

Comments and Suggestions for Authors

Broad summary

The Authors investigate through the Principal Component Analysis how the real devaluation that followed the Great Recession, consisting of both lower employment and substantial wage cuts, affected multidimensional well-being. Results show that the crisis differently hit the various dimensions of well-being as well as the various countries.

Specifically, the Great Recession seems to have changed the priorities in the ranking of the difficulties encountered by European households. The contours of the disadvantage in 2007, before the crisis, mainly refer to deprivation, which might be caused by difficulties on the labor market, while in 2012 the main weakness is found in troubles in the labor market, which may help in becoming poor.

The analysis suggests that the countries with both most valuable economic structure and socio-economic indicators – the western continental and the Nordic countries - have shown a remarkable resilience during the period following the financial crisis, while the Central East countries convergence has stopped and the Southern countries have further been let behind.

Major comments

The paper provides an interesting statistical exercise that highlights the multidimensionality of the real devaluation caused by the Great Recession. There are, however, several aspects of the manuscript and of the empirical analysis that may require further improvements. Specifically:

- In the Introduction, even if the Authors limit their scope to the analysis of the period following the Great Recession, as they quote the EU2020 strategy and its partial failure (lines 108-110), they may report the results of the monitoring activities developed by numerous authors after 2020.

- at the end of the Introduction (before Line 111), Authors are invited to state in few lines the research gap and their research goal (there is a proposition in the abstract that may fit the scope - "Our paper investigates how the real devaluation affected multidimensional well-being through the Principal Component Analysis").

- Concerning their empirical analysis, Authors are invited to explore the availability for 2017 of the same data used for 2007 and 2012, as the real devaluation had persistent effects in many EU countries (clearly, after the COVID-19 there is a structural break).

- Provided that I have some doubts on running a PCA with such a limited number of observations (only 26 countries) with dichotomous variables, Authors are invited to illustrate the statistical reasons behind this choice, and why they have not simply normalized the variables collected.

- Considering that the first two components explain less that 50% of the cumulative variance in 2012, Authors are invited to reconsider (or to motivate) the representation of countries in a two-dimensional scatterplot diagram, as that representation could be severely biased.

- In the Conclusions, Authors are invited to add few lines for each of the following issues: i) research limitations; ii) policy and social implications; iii) suggestion for further research.

Minor comments

As Table 5 and 6 lack of a specific comments, they could be moved to a dedicated Appendix.

At Line 501 there is a typo ("let" instead of "left") 

Author Response

The Introduction has been improved

References have been added

The research design is now hopefully more focussed.

Conclusions have been expanded.

Moreover, point by point:

  • In the Introduction, even if the Authors limit their scope to the analysis of the period following the Great Recession, as they quote the EU2020 strategy and its partial failure (lines 108-110), they may report the results of the monitoring activities developed by numerous authors after 2020 Yes. The major failure was on poverty and social exclusion (Grimaccia 2021) probably also due to the pandemia. Still, we are interested in comparisons before and after the crisis, rather than in the EU2020 achievements. 
  • at the end of the Introduction (before Line 111), Authors are invited to state in few lines the research gap and their research goal (there is a proposition in the abstract that may fit the scope - "Our paper investigates how the real devaluation affected multidimensional well-being through the Principal Component Analysis"). Done.
  • Concerning their empirical analysis, Authors are invited to explore the availability for 2017 of the same data used for 2007 and 2012, as the real devaluation had persistent effects in many EU countries (clearly, after the COVID-19 there is a structural break). EuSIlc issued ad hoc modules for Housing in 2007 and 2012, only.  The ad hoc module  issued in 2018 is only similar, because it does not include all the same variables. Our paper hints at 2018, but the comparability is limited also because the variable AROPE substitutes AROP in the EuSilc 2018 issue.
  • Provided that I have some doubts on running a PCA with such a limited number of observations (only 26 countries) with dichotomous variables, Authors are invited to illustrate the statistical reasons behind this choice, and why they have not simply normalized the variables collected. Our paper is based on the 500k entries supplied by EuSilc micro data. While the countries are 26 only, for each country we rely upon thousands of observations. Yes, we could have simply normalized the variables. Still, we thought that dichotomous variables would sharpen the meaning of the answers (just as how material deprivation is constructed by EuSilc).
  • Considering that the first two components explain less that 50% of the cumulative variance in 2012, Authors are invited to reconsider (or to motivate) the representation of countries in a two-dimensional scatterplot diagram, as that representation could be severely biased. Yes, a warning has been added for 2012 and the scatterplot diagrams were moved to the Appendix. A three dimensions diagram would be less comparable with 2007 (our paper is about comparing these two years) and its visualization is hardly  reader-friendly. 

 In the Conclusions, Authors are invited to add few lines for each of the following issues: i) research limitations; ii) policy and social implications; iii) suggestion for further research. Text has been changed.

As Table 5 and 6 lack of a specific comments, they could be moved to a dedicated Appendix. Done.

At Line 501 there is a typo ("let" instead of "left") Done.

Reviewer 2 Report

Comments and Suggestions for Authors

The paper presents original research that adds new insight to the field. However, I would still suggest the following considerations to further improve the document.

1) The abstract should be more concise and synthesize the key points of the paper. The abstract should start with a contextualisation of the idea. Also, the authors should add a few sentences about the methodology applied and the main results.

2) In the Introduction authors present very broad information about the Great Depression impact on the labour market and income inequality. Still, it does not adequately address the relevance of the study, nor it provides sufficient background information on the main idea of the research. Authors should provide as well the aim of the research and analysis of their main idea. Also, I would suggest authors not only report the previous research but emphasize what was done and what's lacking in the scientific literature which could support the relevance of the research.

3) The literature review section is missing. I would suggest the authors move some information from Section 2 and develop the literature review section explaining the main idea and present argumentation for the relationship between the Great Depression and well-being. Also identifying why authors choose to add two more dimensions to multidimensional well-being. 

3) The paper doesn't have a presentation of methods. the authors only mention The Principal Component Analysis (PCA) but they do not present how it is applied in their research. and in the Estimation result section, the authors use correlation but they do not explain how this method and for what purpose is applied in their research. The authors also use some tests but these are also not explained in the methodology. The cluster detection method also is not explained in the methodology section. 

4) The results could be supported with deeper interpretation and comparison with previous research results

Some minor comments: explain the abbreviation of PCA in Line 187 when it is used for the first time.

Author Response

The Introduction has been improved

References have been added

The research design is now more focussed.

The description of the method has been expanded.

The results have been clarified.

Conclusions have been expanded.

Moreover, by single points:

1) The abstract should be more concise and synthesize the key points … contextualization of the idea. … add few sentences about the methodology and the results. Abstract changed.

2) In the Introduction … not adequately address the relevance of the study, nor it provides sufficient backgroud information on the main idea … what is done and what is lacking in the scientific literature … In the introduction these points were added.

3) The literature review is missing. This section was added. Great Depression is not Great Recession… move information from section 2 yes, done … also explain why authors choose to add two more dimensions. The discussion about the number of dimensions is still open (e.g. the literature on the HDI). Their number may emerge from the chosen methodology (e.g. Betti 2016) or be imposed by a choice of the researcher, which is what we did. 

3) …how PCA is applied Has been expanded

4) The results could be supported with deeper interpretation and comparison with previous research results we added some references

  • Line 187 Done

Reviewer 3 Report

Comments and Suggestions for Authors

The paper is a rigorous study on poverty, standard of living, and well-being in European countries before and after the Great Recession. The authors discuss the topic of well-being using the Eurostat's (European Commission's) and OECD's methodology and use the data for the European Economic Area (EEA) countries, including the UK (pre-2020 composition of the EU). Given the "multidimensionality" of the examined concepts, the authors use the PCA analysis to identify the relevant components and cluster the results to map the cross-country differences.

While the paper is econometrically sound and offers relevant results, the literature review, methodology, and discussion should be further improved before it can be published:

  1. The author(s) provide(s) a comprehensive literature review and discussion of the well-being measurement in the EU countries but omit(s) any studies that analyze the factors' relevance and causality, primarily employing the PCA analysis. Hence, there are no studies to compare the results with/to in the discussion section. Therefore, it is difficult to understand whether the paper contributes new information to the topic.
  2. Most of the methodology related to the indicators is described in the results section, leaving the one on methodology lacking relevant content. This paper is methodology-intensive; hence, the author(s) should be bold in prolonging the methodology section.
  3. The paper's title stresses the EU countries, but the results include the countries of the European Economic Area = EU + Norway, Iceland, and Lichtenstein. The title should be corrected.
  4. The discussion section lacks references to previous research and a comparison of the author(s)'(s) findings with/to it.

The paper includes several misprints and cases of wrong punctuation.

Comments on the Quality of English Language

Please ask the author(s) to proofread the manuscript since there are misprints present (additional letters, punctuation, etc.).

Author Response

The Introduction has been changed

References have been added

Conclusions have been expanded

Moreover, looking at the points raised by the reviewer :

1) … omit(s) any studies that analyze the factor’s relevance and causality, primarly employing the PCA analysis. Hence there are no studies to compare the results … difficult to understand whether the paper contributes new information. Some methodological literature about how to deal with multidimensionality has been added.

2) … methodology related to the indicators is described in the results section … prolonging the methodology section The rearrangement of the sections has been done.

3) EU EEA => the title should be corrected. Since the paper does not include all EEA, nor all EU countries, we refer to “European countries” and the text explains which countries are included (those for which we found enough data).

4) The discussion section lacks references to previous research … reference to papers with similar issues (e.g. comparing quality of life for years 2007 and 2012 in 30 countries) but different methodology.

Misprints and punctuation have been revised

Round 2

Reviewer 1 Report

Comments and Suggestions for Authors

I would like to thank the authors for providing a comprehensive and convincing response to all the questions posed and for updating the manuscript following the indications received. I have no other comments that would help improve the quality of the manuscript.

Reviewer 2 Report

Comments and Suggestions for Authors

Thank you for all the revisions and improvements. I would still suggest adding few sentences in the Abstract about applied methodology and main results.